# Don't call it *privacy-preserving* or *human-centric* pose estimation if you don't measure privacy

**Michele Baldassini*   Francesco Pistolesi*   Beatrice Lazzerini**
Department of Information Engineering
University of Pisa
Largo Lazzarino, 1 – 56122 Pisa (IT)
michele.baldassini@ing.unipi.it, {francesco.pistolesi, beatrice.lazzerini}@unipi.it

## Abstract

This position paper argues that human pose estimation (HPE) cannot be considered privacy-preserving or human-centric unless privacy is measured and evaluated. Although privacy concerns have become more visible in recent years, HPE systems are still assessed almost exclusively using *accuracy* metrics. Privacy is neither defined in measurable terms nor linked to regulatory requirements, and common deployment architectures introduce additional risks due to data transmission and storage. We highlight the limitations of current practices, including the continued reliance on RGB inputs and the lack of benchmarks that reflect legal and ethical constraints. We call for a shift in evaluation practices: *privacy* must become part of how HPE systems are designed, tested, and compared.

## 1   Introduction

**Position: This paper argues that human pose estimation must shift from performance-only evaluation to human-centric approaches that measure and prioritize privacy.**

Human pose estimation (HPE) determines the position of human joints and connections from images or videos. It has achieved impressive results in the last decade, becoming key in computer vision with applications to human-computer interaction [1], surveillance [2], entertainment [3], sports analytics [4], and healthcare [5]. These advances have been facilitated by powerful deep learning architectures [6] and large-scale, richly annotated datasets containing RGB images or videos collected in ideal [7–9] or synthetic [10–12] settings. Recent years have seen growing concerns about privacy, as AI models can expose *sensitive personal information (SPI)*, for example, facial features, body morphology, gender, or ethnicity [13]. This led to exploring anonymization methods and data types that capture less sensitive information than RGB images, such as LiDAR scans, thermal images, and depth maps.

Real-world HPE applications, including workplace safety, ergonomic risk assessment, physical rehabilitation, and elderly care, require the deployment of HPE systems in environments where privacy, regulation, and practical constraints are mandatory. These domains involve detailed, continuous monitoring of human movement, and have promoted datasets such as *IKEA ASM* [14] (multi-view recordings of furniture assembly tasks), *UCO-Labeled* [15] (physical exercises), and *MMRI* [16] (multimodal rehabilitation data). These datasets assume that accurate pose estimation requires detailed visual input, often revealing SPI.

This assumption is problematic, as privacy protection is a legal, ethical, and social obligation in real-world contexts. In particular, regulations require data minimization, purpose limitation, and explicit consent, thus making the acquisition of sensitive image data incompatible with these principles.

---

*Equal contribution.

39th Conference on Neural Information Processing Systems (NeurIPS 2025) Position Paper Track.

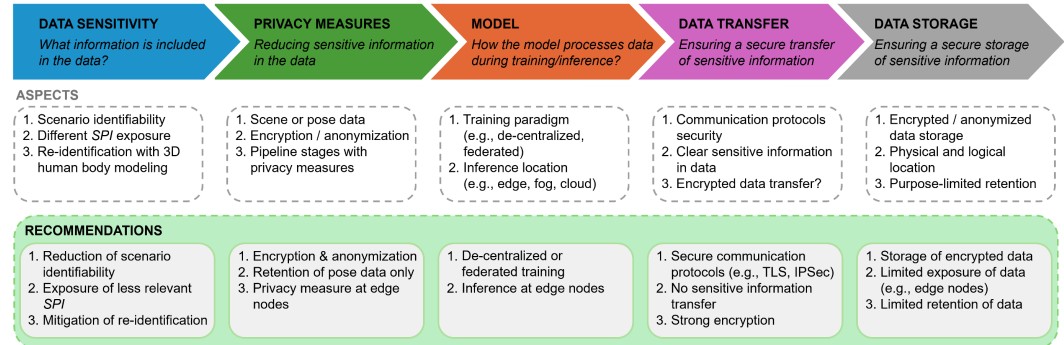

Figure 1: Overview of aspects and recommendations related to some key dimensions for privacy assessment.

Privacy concerns also involve the representation models of the body used in HPE. For example, the *kinematic model* only provides joint coordinates and is relatively privacy-preserving; the *planar model*, based on silhouettes or statistical shapes, reveals more about body posture and form; the *volumetric model*, used in 3D mesh reconstruction, encodes body shape and movement with enough fidelity to infer gender, age, or health conditions, thus posing serious privacy concerns.

Many modern HPE systems achieve high accuracy, but continue to use RGB data, which contain highly identifiable features [17–19]. This leads to serious privacy risks, including re-identification and unauthorized use. In addition, real-world HPE system deployments rely on *distributed architectures* that process data across multiple layers to distribute the computational load. Privacy risks are thus increased, as sensitive data are transferred across multiple layers—for example, from local acquisition and inference (edge) to centralized aggregation or training (cloud). Protecting sensitive data across system layers requires long-term privacy measures. HPE systems have started to address these challenges using techniques such as obfuscation, alternative data types (e.g., Wi-Fi, thermal, LiDAR), secure transmission and computation (e.g., TLS, encryption), and decentralized training (e.g., federated learning). These methods reduce privacy risks while supporting accuracy and legal compliance. Still, balancing privacy, performance, and deployment constraints remains a challenge.

Although privacy has received more attention in recent HPE systems, their performance is only measured using precision indicators for 2D HPE [20, 21] and 3D HPE [22–27]. Privacy is evaluated without using metrics or referring to regulatory standards, while also overlooking the risks due to data transit, storage, and processing when deploying HPE systems on multi-tier architectures, e.g., *edge-fog-cloud* [28]. Figure 1 shows, from left to right, some key dimensions that characterize many modern HPE systems, the aspects to consider for each dimension, and some recommendations to address to increase the level of privacy.

These limitations highlight a gap between current research practices and the need for privacy-aware, real-world applications. Addressing this gap requires defining privacy as a measurable aspect of HPE, and making it part of the system evaluation.

We invite the NeurIPS community to play an active role in redefining the foundations of HPE. We do not lack technical capabilities, but we lack standardized benchmarks, community incentives, and a shared vision grounded in real deployment contexts. We outline the limitations of current approaches, identifying key ethical and regulatory gaps. Finally, we reflect on how to design, develop, and deploy human-centric datasets and models that combine performance and privacy.

The paper is structured as follows: Section 2 explores alternative solutions to preserve privacy in HPE; Section 3 presents a view on possible ways for measuring the privacy risk when designing and deploying HPE systems; Section 4 discusses some alternative views; Section 5 concludes the paper.

## 2    How to preserve privacy?

The shift toward privacy-preserving modalities and system architectures often entails a trade-off in positional accuracy—particularly in the localization of fine-grained joint coordinates. Privacy-driven

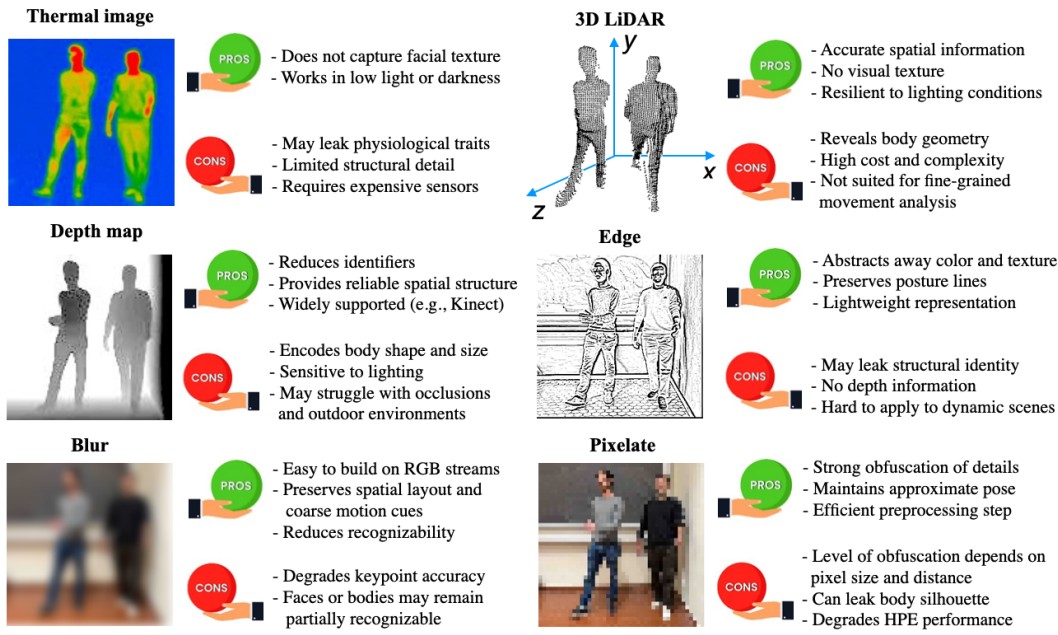

Figure 2: Examples of data modalities used in privacy-aware pose estimation, each with some pros and cons in terms of accuracy and privacy.

obfuscation, reduced resolution, and sensor substitution (e.g., depth in place of RGB) can significantly degrade the precision of joint-level estimations. This, in turn, affects downstream computations such as joint angle estimation, body segment alignment, and movement smoothness—metrics that are essential in domains governed by regulatory standards.

## 2.1 Changing modality

While RGB-based datasets dominate the field for their high accuracy, their inherent identifiability raises significant ethical and legal concerns. Conversely, alternative modalities (depth, LiDAR, thermal, RF) offer privacy-preserving properties but vary widely in usability, generalizability, and task-specific performance. This section argues that no single modality is universally "best" for privacy; instead, the field urgently requires a standardized measurement framework to evaluate trade-offs across modalities systematically.

Recent HPE datasets include privacy-preserving modalities—such as depth, LiDAR, thermal, and RF—often alongside RGB (see Figure 2). Although multimodal datasets such as NTU-RGBD [29,30] and LiDARHuman26M [31] show high performance with non-visual data, RGB datasets, e.g., COCO [11], continue to dominate due to their accuracy, despite high identifiability risks.

Synthetic datasets (e.g., SURREAL [32]) and RF/thermal datasets (e.g., MM-FI [33]) offer promising privacy advantages, yet lack scale, realism, or consistent evaluation. High-precision datasets like Human3.6M [34] provide detailed annotations but amplify privacy concerns.

Non-visual sensing modalities, such as radio frequency (RF), depth, and thermal imaging, are often presented as privacy-preserving alternatives to RGB. However, these technologies maintain sensitive personal information, and there is currently no standard metric to compare their privacy levels.

**Radio frequency (RF).** RF-based HPE methods—including RFID, FMCW radar, mmWave radar, and Wi-Fi—enable pose detection without visual data, providing anonymity [35–47]. While avoiding facial or direct visual biometric capture, these systems extract detailed skeletal and movement patterns that can act as unique biometric identifiers. This creates privacy risks related to covert tracking, re-identification, and behavioral profiling, especially in private or sensitive environments. Recent advances improve pose detail, increasing the sensitivity of collected biometric data and emphasizing the need for strong privacy safeguards.

Table 1: Modalities with associated pose representations and privacy risks.

| Modality | Pose Representation | Privacy Risks |
|---|---|---|
| RGB images | 2D skeletons full-body meshes | High identifiability; re-identification via appearance cues, clothing, background context; face recognition leakage. Dominates accuracy but raises ethical/legal concerns [11, 29, 30]. |
| RF Wi-Fi | 2D/3D skeletons motion trajectories | Skeletal/motion info can reveal unique biometric signatures [94]; covert tracking [95]; re-identification; behavioral profiling [96]; environmental noise does not fully prevent re-identification [48–50]. |
| Depth sensing | 2D/3D skeletons voxel grids meshes | Body-shape re-identification; motion patterns and behavioral profiling; fusion with RGB data increases privacy risks [59–61, 73, 74]. |
| Thermal sensing | 2D/3D skeletons meshes | Body-shape re-identification; motion patterns, behavioral profiling and physiological traits (e.g., emotion and temperature); fusion with other modalities increases risk [83–85, 88, 89]. |

**Wi-Fi.** Wi-Fi-based HPE uses reflections of standard Wi-Fi signals to infer body poses without visual cameras [48–58]. However, despite the environmental factors, signal noise, and lower spatial resolution compared to radar systems, Wi-Fi HPE can still capture detailed skeletal and motion information that reveal unique biometric patterns. Therefore, strong privacy safeguards are essential to prevent misuse of pose data and protect individuals' anonymity in Wi-Fi sensing applications.

**Depth sensing.** Depth map-based HPE reconstructs 3D body poses without RGB images [59–76], yet the detailed skeletal data it generates can serve as biometric identifiers, enabling person recognition and behavioral profiling. Fusion with RGB data further elevates privacy risks [73, 74].

LiDAR-based HPE provides high-resolution 3D body reconstructions [77–82], capturing uniquely identifiable body shapes and motions. Its growing use in the public and medical domains raises concerns about covert tracking and unauthorized surveillance.

Both modalities extract biometric signatures from non-visual data, necessitating strict privacy protections to prevent misuse and ensure anonymity.

**Thermal sensing.** Thermal imaging enables HPE without visible light, offering some privacy benefits [83]. However, it still reveals body shapes and motion patterns that can serve as biometric identifiers, risking re-identification. Additionally, thermal data expose physiological traits such as emotional states and environmental responses [84, 85], raising further privacy concerns.

Although lower in resolution, thermal images combined with depth data improve accuracy, but increase the risk of exposing identifiable features [86, 87]. Adaptations of visible-light pose models to thermal imaging [88–92] confirm persistent risks of biometric and physiological data leakage. Large annotated datasets [88, 89, 93] facilitate development but also highlight the need for careful privacy protection.

Thermal imaging partially protects visual identity, but still reveals sensitive biometric and physiological information, requiring strict privacy measures.

Table 1 summarizes the sensing modalities, their pose representations, and their privacy risks.

## 2.2 Anonymize visual data

Reducing the amount of identifiable information in visual data is an easy way to improve privacy in HPE systems. However, anonymization is not a guarantee. Each technique has trade-offs, and many preserve residual cues that can still identify individuals.

Lowering image resolution hides facial features and fine details, but often leaves body shape, gait, and motion patterns intact. These structural signals are enough to re-identify subjects in many cases, especially when combined with temporal information [97–99].

Software-based obfuscation techniques such as blurring or noise injection (for example, by using pixelation) are easy to apply and effective at masking sensitive visual content (see the bottom line of Figure 2). However, they degrade the quality of keypoint detection, thereby reducing accuracy. Also,

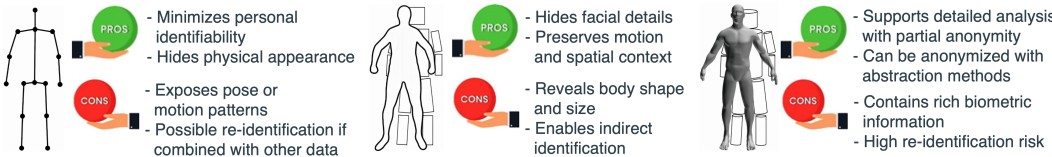

Figure 3: Comparison of *kinematic*, *planar*, and *volumetric* human pose representations (from left to right), showing the relation between abstraction level, identifiable information, and privacy risks.

obfuscated data are not safe by default: several studies show that neural models can reconstruct or infer masked information using priors or multi-frame context [100–102].

Hardware-based alternatives, such as event cameras, infrared sensors, or low-resolution depth cameras, limit the amount of biometric information captured at the source [103–105]. This can reduce the need for obfuscation, but has limitations including cost, and residual physiological traces—for example, heat signatures or breathing motion—that can still leak private information.

Hybrid setups exist that combine the strengths of both approaches. For example, some apply optical filters or hardware constraints during acquisition, followed by software processing [106, 107]. Although these hybrid techniques reduce direct exposure, they still require access to partially informative raw data at some stage in the pipeline, which reintroduces privacy risks.

Techniques like federated learning can help reduce data exposure by keeping raw data close to the acquisition point [108]. But again, this does not guarantee privacy: model updates can leak sensitive features, and attacks based on gradients or reconstruction from weights remain an open concern.

Thus, anonymization is not a solution. Systems that are claimed to preserve privacy should prove which types of personal information are still accessible after processing, and under what conditions.

### 2.3 Adopting privacy-aware pose representations

The way we represent human pose has important consequences for privacy. Different model types expose specific levels of personal information.

For example, kinematic models, often used in both 2D and 3D pose estimation, represent the body as a set of joint positions connected by limbs. These models focus on structure and motion, not appearance, making them relatively privacy-friendly. Planar models, which use silhouettes or simplified contours, provide more visual information about posture and body shape [109]. This can reveal physical traits that may enable identification or profiling. Volumetric models generate full-body 3D meshes with shape and motion details [110–112]. These representations can encode sensitive attributes such as gender, age, or physical condition, even without RGB input. Figure 3 shows the advantages and privacy risks of each model based on the level of abstraction and identifiable information retained.

Choosing a pose representation is thus both a technical and a privacy decision. We should consider what level of detail is needed for the task at hand, and whether a simpler representation could meet the goal while better protecting privacy.

Thus, pose estimation is privacy-preserving only if we can measure privacy and consider it at every stage: data lifecycle, model training and evaluation, and final deployment.

## 3 How can we start measuring privacy?

This section gives some directions for evaluating privacy in HPE. Starting from legal requirements, we give an example of how each part of an HPE system (data, model, inference, and transmission) contributes to the overall privacy risk.

### 3.1 Privacy risk factors

Privacy risk in HPE depends on various design choices, including:

- the type of *scene data* collected and how *human pose* is represented;

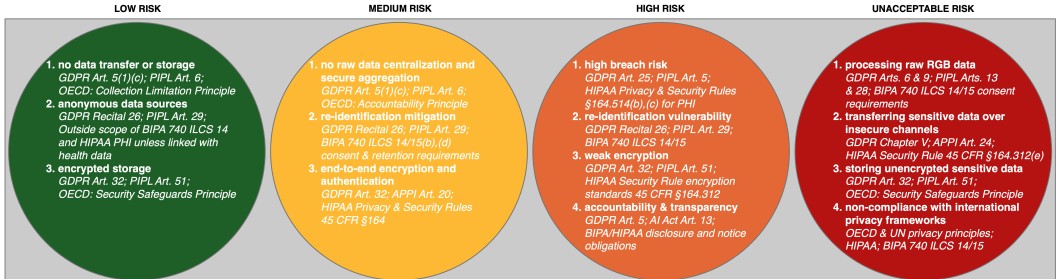

Figure 4: Risk levels in HPE systems and examples of privacy practices. Each circle represents a category of privacy exposure ranging from low (green) to unacceptable (red), with legal references from GDPR, PIPL, APPI, and BIPA.

- where and how *inference* is performed;
- where and how *training* is performed, and how model updates are distributed;
- where the *storage* takes place, which data are stored and for how long;
- how data *transmission* occurs and how many devices are involved in transfer and storage.

For example, a model trained on local data and never exposed to raw video is safer than one trained on images stored remotely. If personal data are kept longer than needed, or reused for other purposes, the risk is increased.

## 3.2   A risk-based view based on regulations

Legal frameworks such as the *General Data Protection Regulation (GDPR)* , the *AI Act*, the *Personal Information Protection Law (PIPL)*, and the *Biometric Information Privacy Act (BIPA)* part of the *Illinois Compiled Statutes (ILCS)* provide clear principles to handle personal and biometric data. These include data minimization, purpose limitation, storage limitation (GDPR Art. 5), privacy by design and by default (Art. 25), and restrictions on the use of biometric data (Art. 9). The AI Act defines high-risk systems (Annex III), including those used for biometric categorization or workplace monitoring.

Similar rules appear in China's PIPL (Art. 28 and 51) and Japan's *Act on the Protection of Personal Information (APPI)* (Art. 23 and 24), which also limit data retention and international transfers.

In the U.S., the BIPA imposes strict requirements on the collection and storage of biometric identifiers (ILCS chapter 740 Act. 14/15). Also, the *Health Insurance Portability and Accountability Act (HIPAA)* establishes standards for the protection, use, and disclosure of any identifiable health data (*protected health information (PHI)*) handled by covered entities and their associates.

At an international level, the *Organization for Economic Co-operation and Development (OECD)* and *United Nations (UN)* guidelines promote principles such as accountability, transparency, and proportionality in AI and data processing.

Due to their widespread adoption worldwide, we should use these principles as a starting point to evaluate the privacy risk of HPE systems in years to come.

We try to do this by outlining four levels of risk (*low*, *medium*, *high*, and *unacceptable*), and describe how typical design choices map to them. Our goal is not to propose a metric, but to show that one way exists to start measuring the privacy risk, and it could be based on regulatory frameworks already adopted globally.

**Low Risk.**   Inference is performed locally on the device. The input is a non-identifiable data type (e.g., body keypoints, silhouettes, or RF signals). Data are not stored or are stored in encrypted form, only for short periods. Transmission is absent or encrypted. These setups follow the principles of data minimization and privacy by design (GDPR Art. 5, 25), and reflect OECD guidelines such as *Collection Limitation* and *Security Safeguards*. These systems typically fall outside the scope of regulated biometric data—such as BIPA or HIPAA, unless linked with health records.

**Medium Risk.** Input data include structured but potentially re-identifiable information (e.g., depth maps or thermal images). Inference is local, but training or aggregation may occur on external servers. Short-term storage or encrypted transfer is present. These setups may comply with regulation, but require documentation and justification under GDPR (e.g., Art. 6 on a lawful basis). As depth and thermal data may enable biometric profiling, it is crucial to comply with regulatory regimes such as the BIPA with consent and retention rules (e.g., 740 ILCS 14/15(b), (d)), HIPAA's PHI safeguards under the *Privacy* and *Security Rules*, and OECD's principles—including use limitation and accountability.

**High Risk.** RGB video is collected and stored. Inference may be offloaded to cloud services. Data are transmitted without strong encryption or stored long-term. The system is used in sensitive contexts such as health or workplace monitoring. This configuration may involve biometric data (GDPR Art. 9) and fall under high-risk use cases in the AI Act (Annex III), requiring a full risk management process. In the U.S., health-linked biometric data become PHI under HIPAA's Privacy/Security Rules (including de-identification rules under §164.514(b), (c) and Safe Harbor/Expert Determination methods), and BIPA's strict notice, consent, and retention obligations (740 ILCS 14/15) also apply.

**Unacceptable Risk.** Biometric data are collected without consent. RGB or 3D mesh data are stored and reused for purposes unrelated to the original intent. Data are processed in jurisdictions without regulatory protection or oversight. These configurations likely violate GDPR and PIPL, BIPA's consent and disclosure requirements (e.g., 740 ILCS 14/15), HIPAA privacy safeguards, OECD/UN principles, and would be non-compliant even under basic legal scrutiny.

Figure 4 summarizes examples of practices in HPE, grouped by privacy risk level and aligned with relevant legal provisions. This classification does not require any new regulation. It builds directly on existing law and links technical decisions in HPE to levels of risk. These levels could be part of benchmark reporting and accuracy metrics to support fair comparisons between systems.

### 3.3 Toward practical privacy indicators

The risk levels discussed in the previous section can help assess the privacy exposure of HPE systems, but are insufficient for comparing or evaluating models in practice. Unlike accuracy, a privacy score is never reported in benchmarks or model documentation. This makes it difficult to assess the trade-off between performance and privacy or to understand how a system handles personal data.

We argue that it is possible to introduce basic privacy indicators—simple, interpretable, and aligned with legal principles—that can be used during system design and evaluation. These indicators do not aim to capture all aspects of privacy, but they can make privacy visible and comparable. Some possible approaches, which could also be combined with each other, may be based on the ideas as follows. These examples show that privacy in HPE can be evaluated in multiple ways, using system-level properties and practical indicators. None is definitive, but together they could support benchmarks where privacy is not ignored, but reported and considered alongside performance.

#### 3.3.1 Scoring input modalities

The input data used by an HPE system directly affect the amount of SPI exposed. Some modalities, such as keypoints or RF signals, contain little or no identifiable information. Others, like high-resolution RGB with visible faces and background, capture a wide range of biometric and contextual features. A scoring scheme could assign a privacy exposure value to each modality. For example:

- 0 points: keypoints, RF signals, IMUs;
- 1 point: silhouettes, depth maps, thermal images;
- 2 points: obfuscated or low-res RGB;
- 3 points: high-res RGB with visible face and background.

This type of score reflects the level of exposure before any processing, and could be extended to systems using multiple modalities.

### 3.3.2 Penalizing risky handling of data

Privacy is not determined by the input alone. It also depends on what happens to the data after they are captured. Design choices such as where inference occurs, how data are transmitted, and whether they are stored introduce additional levels of risk. One way to consider this aspect could be to design a penalty system that assigns one point to each design choice that increases privacy risk:

- cloud-based inference;
- unencrypted data transfer;
- long-term storage;
- reuse of data for different purposes.

This cumulative score would reflect how many risk factors characterize the pipeline.

### 3.3.3 Privacy labels based on legal categories

Instead of numeric scores, an HPE system could carry a structured label that considers privacy-relevant properties, inspired by GDPR and the AI Act (but also other frameworks). For example:

- uses biometric data (GDPR Art. 9): yes/no;
- AI Act risk level: minimal / limited / high;
- data retention: none / short / long;
- international transfer: none / with safeguards / unrestricted.

This label does not measure privacy but highlights legal exposure and promotes early consideration of regulatory constraints.

### 3.3.4 Re-identification resistance

A more empirical option could be based on measuring the extent to which a system's output can help re-identify individuals. This option could use, for example, a re-identification classifier to test whether blurred images, silhouettes, or depth maps still reveal identity.

Recent studies have shown that 2D/3D skeletons and 3D meshes can support human re-identification. In particular, skeleton-guided feature learning enables recognition across significant visual variations [113, 114], whereas mesh-based modeling highlights the potential of structured geometric outputs to preserve discriminative cues for re-identification in complex scenarios [115]. Various anonymization strategies have been proposed to mitigate this risk. For example, motion retargeting and adversary-guided retargeting which, respectively, anonymize skeleton data while preserving realistic motion patterns [116], and reduce the re-identification risk without degrading action utility [117]. Also, 3D human pose and shape reconstruction have demonstrated that structured outputs facilitate re-identification, highlighting the need for privacy-preserving mechanisms [118].

Beyond pose data, visual privacy scores derived from image attributes provide an empirical framework for quantifying privacy risks [119]. In addition, self-supervised learning can suppress private information at the feature level [120].

Re-identification resistance could thus help generate a privacy score based on actual model behavior rather than assumptions. It could be used alongside accuracy, latency, and other performance metrics to compare systems more fairly in privacy-sensitive settings.

### 3.4 Performance vs Privacy

Many HPE methods that improve privacy tend to reduce accuracy as a side effect. For example, using depth maps, thermal images, or radio frequency data instead of RGB often decreases keypoint detection quality. Local inference limits the model size and capacity. Obfuscation techniques, such as blurring or pixelation, can hide identifiable traits but degrade spatial precision. These trade-offs are well known, but rarely measured or reported explicitly.

Existing benchmarks typically evaluate models on clean RGB input in ideal conditions. As a result, models optimized for accuracy may perform poorly when introducing privacy constraints.

Systems designed for real-world deployment—such as rehabilitation, workplace safety, or smart environments—must be tested under realistic assumptions about sensing, storage, and processing.

We suggest that benchmarks should include both accuracy and privacy risk, based on the levels described in Section 3. For example, a model could be reported as achieving 85% AP@0.5 under a medium privacy risk configuration, or 78% under a low-risk configuration. This would make trade-offs clear and comparable.

Various directions may help reduce the gap between privacy and performance. These include designing models that are robust to low-information inputs (e.g., edge-based features, coarse silhouettes), fine-tuning on privacy-filtered datasets, and using modality fusion (e.g., combining IMU and depth). Also, privacy-aware training objectives can be explored. Lightweight architectures and real-time inference pipelines can further support local and secure deployment.

Privacy and accuracy are not mutually exclusive, but optimizing for both requires shifting how we design, train, and evaluate HPE systems.

# 4  Alternative views

Despite growing attention to privacy, much of the current HPE research is shaped by assumptions that sideline or oversimplify it. These assumptions often stem from choices in datasets, benchmarks, or evaluation protocols, where privacy is either abstracted away or treated as an afterthought. Below, we highlight alternative perspectives that continue to inform the field and which we believe should be critically examined.

**Accuracy is the main objective.** Most HPE models are designed and evaluated based on accuracy scores such as PCK, MPJPE, or AP [121–123]. This view only considers privacy when it comes to deploying models, not during development or benchmarking. This underestimates the role of upstream data flows and model behavior.

This approach overlooks real-world scenarios where privacy is a legal and ethical requirement from the very beginning. A highly accurate model that cannot be deployed due to privacy risks has a limited practical value.

**Switching to non-RGB data guarantees privacy.** Some researchers assume that using depth maps, LiDAR scans, thermal images, or RF signals is enough to guarantee privacy. HPE systems that rely on these inputs are described as "privacy-preserving" [124–126].

However, this is overstated, irrespective of the data modality. In general, non-RGB data can reveal body shape, walking style, or even signs of stress or illness. What matters is what the model can still learn or infer, rather than what the image looks like.

**Privacy is difficult to measure.** Another common belief is that privacy is too vague or context-specific to define, and that is why it is ignored [127–130]. However, some regulations define several aspects of privacy, although they do not give a universal definition of privacy.

In the previous section, we proposed dimensions along which we can assess privacy risks in distributed HPE systems. These dimensions are grounded in international regulations and include the type of data used, how the model is trained, where inference takes place, and how data are stored.

We thus believe that privacy is not impossible to measure, but requires clear and shared criteria to comply with international regulations.

**Synthetic or augmented data preserve privacy.** Some works propose generating synthetic datasets [131–133] or augmenting real datasets with generative models as a privacy workaround. This approach reduces reliance on real-world data, but introduces other risks. For example, synthetic data can still replicate sensitive patterns from training input (e.g., via memorization in GANs), and augmented data may retain identifiable traces if based on insufficiently anonymized inputs. Risk should be evaluated based on what can be reconstructed or inferred from model output.

**Privacy-preserving means not storing or transferring data.** Some system architectures are often claimed to be privacy-preserving only because they do not retain input data after inference [134–136]. Although data retention is crucial to assess privacy, it is not the only aspect. Real-time systems that process data in memory without logging may still expose personal information through live

streams, insecure APIs, or inference-time side channels. Privacy requires more than avoiding storage, it requires minimizing collection, restricting transmission, and securing processing across all layers.

These views reflect a broader trend in HPE research: privacy is often considered a deployment detail, not a fundamental design concern. We believe this is incompatible with the regulatory, ethical, and societal expectations surrounding real-world AI systems. Reframing privacy as a measurable, multi-dimensional property of system behavior (not just data type or retention policy) is crucial if HPE models are to be trusted, adopted, and legally compliant.

## 5   Conclusions

Human pose estimation (HPE) has reached high levels of accuracy and is ready for deployment in many real-world settings. But if these systems are to be used in workplaces, healthcare, rehabilitation, or public spaces, they must go beyond performance. Accuracy alone is no longer enough.

In this position paper, we have argued that privacy must be treated as a core evaluation criterion in HPE, not an afterthought. We have highlighted how current practices overlook key risk factors—such as data type, storage, and inference location—and how common assumptions, such as *non-RGB equals privacy* fail. And we have outlined how legal frameworks already provide structure for thinking about risk, while also showing that measurable indicators can be introduced into benchmarks to reflect privacy exposure.

Then, we have presented some practical directions for evaluating privacy, including modality scores, penalty-based indicators, legal labeling, and re-identification resistance testing. These tools do not replace regulation, but make privacy visible so that researchers can compare systems by how well they perform and to what extent they respect privacy.

The time has come for a shift. Privacy and trust must be treated as design requirements, and we already have the means to reduce identifiability, secure data, and design models that are compatible with regulation and societal expectations. The next step is to build the benchmarks, tools, and incentives that make privacy a standard part of the progress in HPE.

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
