# OpenReview forum: "Don’t call it privacy-preserving or human-centric pose estimation if you don’t measure privacy"
_NeurIPS.cc/2025/Position_Paper_Track — NeurIPS 2025 Position Paper Track_

### Official Review · Reviewer_rkFY · 2025-08-05

**Significance:** 3
**Presentation:** 3
**Rating:** 7
**Confidence:** 3

**Summary:**

Paper position: human pose estimation (HPE) must shift evaluation from performance-only to one that also includes measuring and prioritising privacy.

The paper argues that real world applications are deployed to environments where privacy, regulation, and practical constraints are mandatory. Yet, many high-accuracy HPE systems come with serious privacy risks. Although privacy has received more attention in recent HPE systems, privacy is evaluated without using metrics or referring to regulatory standards. The proposal here is that the community should work towards defining privacy as a measurable aspect of HPE, making it part of the system evaluation, to mitigate the gap between current research practices and the needs of privacy-aware real-world applications.

The paper outlines ways to preserve privacy, and factors that we can use to measure privacy. Finally, a number of alternative views are identified and argued against, supporting the paper position.

**Strengths:**

The paper gives a thorough assessment of ways to preserve and measure privacy, from practical and regulatory points of view. It discusses the performance vs privacy tradeoff, and argues for including both performance and privacy metrics in evaluation, making tradeoffs clear and comparable. These points clearly support the paper position. References can be found in almost every single argument. Alternative views are update-to-date and practical, and are addressed with reasonable arguments.

**Weaknesses:**

I do not see any major weakness in the paper. It has a strong, practical, and valid view point.

**Questions:**

I do not have any major question. To me, what has been asked here in the paper is just explicit clarity in privacy preserving when it comes to designing, training and evaluating HPE systems. But such extra data can bring about substantial benefits: in comparing and understanding HPE systems under different privacy preserving requirements, and in deploying such systems to regulated places.

**Alternative Position:**

Yes, and alternative positions are well-considered and addressed by the argument

**Author Identification:**

No.

**Context:**

3

**Discussion:**

4

**Ethics:**

["NO or VERY MINOR ethics concerns only"]

**Position:**

Yes, the paper argues for or against a position related to machine learning.

**Support:**

3

**Thoroughness:**

4

---

### Official Review · Reviewer_5P9s · 2025-08-13

**Significance:** 4
**Presentation:** 4
**Rating:** 9
**Confidence:** 4

**Summary:**

This paper argues that human pose estimation (HPE) cannot be considered privacy-preserving or human-centric unless privacy is measured and evaluated. It outlines that frameworks are needed to help developers balance accuracy with privacy concerns.

**Strengths:**

This paper very clearly outlines the challenge with balancing privacy with accuracy in HPE, the existing privacy risk mitigation and why those techniques need to scored/evaluated and not just universally applied. I really enjoyed graphic 2. The paper engages with existing literature and contributes to a really important NeurIPS topic (HPE but also biometrics and medical AI).

**Weaknesses:**

The only section I struggled with was Section 3. The paper just starts referencing GDPR, the AI Act, and the PIPL. There is no explanation of what these laws are, where they are from (EU/China) and more importantly why choose these to inform the risks? The US still (for now) is a leader in AI/medical research so why not include HIPAA or Illinois Biometrics Law? Or why not look for a document from OECD/UN that outlines shared global principles? Just a bit more about these laws and why the authors chose them to frame the risks around them would be helpful.

**Questions:**

see above, I would love some insights into why the authors chose GDPR/PIPL to inform risks as opposed to other frameworks.

**Alternative Position:**

Yes, and alternative positions are well-considered and addressed by the argument

**Author Identification:**

No.

**Context:**

4

**Details Of Ethics Concerns:**

This paper makes an important contribution to the field of CS ethics.

**Discussion:**

4

**Ethics:**

["NO or VERY MINOR ethics concerns only"]

**Position:**

Yes, the paper argues for or against a position related to machine learning.

**Support:**

4

**Thoroughness:**

5

---

### Official Review · Reviewer_pPex · 2025-08-16

**Significance:** 3
**Presentation:** 3
**Rating:** 4
**Confidence:** 4

**Summary:**

This paper holds the position that pose estimation must be more privacy focused and not potentially leak subject information. Towards that end, the paper proposes a few alternatives for human pose estimation that are more privacy preserving. Some of those alternatives are changing the input modalities in which the pose is estimated or change the representation for the human pose to be more privacy preserving. The paper also proposes a framework for determining privacy violations and the possible ramifications of violating it.

**Strengths:**

1. The paper is well written and quite comprehensive from a privacy analysis perspective

2. Some of the risks are quite intuitive to understand

**Weaknesses:**

1. A major weakness of this paper in my opinion is the bundling of privacy breaches at the data input level (which are absolutely not specific to human pose estimation only) along with privacy breaches at the output which are potentially actually leaky

2. I don’t see very strong evidence that the predictions of human pose are actually leaking private information.

**Questions:**

1. How would we prove that private information of subject are actually being leaked by the pose estimation method? There’s a possibility, but unless I missed it (please do correct me!), it hasn’t been proven in the paper.

**Alternative Position:**

No

**Author Identification:**

No.

**Context:**

3

**Discussion:**

3

**Ethics:**

["NO or VERY MINOR ethics concerns only"]

**Position:**

Yes, the paper argues for or against a position related to machine learning.

**Support:**

2

**Thoroughness:**

3

---

### Note · Authors · 2025-09-05

**1-10 Additional Comments:**

The position paper track has been a nice opportunity to present arguments that are conceptual, regulatory, and forward-looking rather than purely experimental. We appreciate that the reviewers engaged with both the ethical and technical aspects of the work. The reviews were highly balanced as they focused on technical aspects, regulatory framing, and the paper overall positioning. This diversity of perspectives was really helpful. Overall, the process has been positive. The feedback we received was constructive and will allow us to clarify the contribution and make the paper more useful to the community. We believe the position paper track is important for shaping research agendas at NeurIPS and are glad to have participated.

**1-11 Submit Again:**

Definitely yes

**1-1 Submission Process:**

5

**1-2 Next Year:**

It would be interesting to see the position paper track continue to highlight contributions that connect machine learning with broader issues such as privacy, ethics, regulation, and deployment. Next year, it could also be valuable to create opportunities for interaction around accepted position papers, for example through dedicated discussion sessions or panels, so that these works can stimulate conversation and exchange of perspectives within the community.

**1-3 Future Development:**

One idea would be to provide dedicated spaces where accepted position papers can directly foster dialogue—for example, through roundtables or joint sessions that bring together authors and audience members. This would amplify the impact of the track by turning position papers into starting points for community debate. Another possible improvement is to make the track highly visible in the final program, so that conceptual and regulatory discussions receive attention alongside technical advances.

**1-4 Interest:**

["Panel discussions with other position paper authors", "Workshops for developing position papers"]

**1-4 Other Interest:**

Short interactive sessions with participants specifically interested in our paper, to allow for focused discussion and exchange of ideas.

**1-5 Thoughtful:**

10

**1-6 Supportive:**

10

**1-7 Technical Aspects Versus Position:**

9

**1-8 Gate Keeping:**

10

**1-9 Camera Ready Changes:**

Based on the reviews received, we will make the camera ready changes as follows:
1) Clarifying privacy leakage: we will refine Section 2 with additional references showing that pose representations can reveal sensitive information. We will also add a summary table linking modalities and pose representations to the corresponding privacy risks.
2) Expand regulations: we will expand Section 3 by adding the frameworks suggested by one of the reviewers (i.e., HIPAA, Illinois BIPA, OECD/UN guidelines) and more context about the legal frameworks we chose (GDPR, AI Act, PIPL).

**3-1 Review Response1:**

5P9s

**3-2 Reaction To Review1:**

This review was very thoughtful and strongly supportive of the paper position. The reviewer appreciated our work and clearly understood the core message. The feedback was very constructive, pointing out that Section 3 would benefit from more context about the legal frameworks we chose (GDPR, AI Act, PIPL) and suggesting expansion to include U.S. or global regulations. This is a valuable recommendation that we will follow to strengthen the paper.

**3-3 Review Response2:**

rkFY

**3-4 Reaction To Review2:**

This review was thoughtful and supportive of the position taken by the paper. The reviewer highlighted that the paper presents a strong, practical, and valid viewpoint, highlighting that the paper contains references for almost every argument. The focus was balanced between technical aspects and the position itself. The tone was constructive, fair, and aligned with the paper goals. The reviewer did not identify any major weaknesses.

**3-5 Review Response3:**

pPex

**3-6 Reaction To Review3:**

The reviewer highlighted that the paper is well written and the tone of the review was constructive and respectful. We noted that, differently from the other two reviewers, this reviewer marked ‘No’ under “Alternative Positions.” That was certainly accidental as the paper contains a dedicated section, “Section 4: Alternative views” (on page 8), as explicitly requested by the call. The reviewer raised one question (referred to as Q1) that we report below, with our response (R1).

Q1: “How would we prove that private information of subject are actually being leaked by the pose estimation method? There’s a possibility, but unless I missed it (please do correct me!), it hasn’t been proven in the paper.”

R1: The paper already cites works showing the risk of privacy leakage when using HPE systems (references [17]-[19] on page 2). To add evidence that privacy information of subjects can actually be leaked by pose estimation methods we will add more references that explicitly prove this risk and add a summary table linking pose representations to the corresponding privacy risks reported in the literature. Key references we will add include those as follows:

[A] J. Liu, B. Ni, Y. Yan, P. Zhou, S. Cheng and J. Hu, "Pose Transferrable Person Re-identification," 2018 IEEE/CVF Conference on Computer Vision and Pattern Recognition, pp. 4099-4108, 2018.
[B] C. Song, Y. Huang, W. Ouyang and L. Wang, "Mask-Guided Contrastive Attention Model for Person Re-identification," 2018 IEEE/CVF Conference on Computer Vision and Pattern Recognition, pp. 1179-1188, 2018.
[C] J. Chen et al., "Learning 3D Shape Feature for Texture-insensitive Person Re-identification," 2021 IEEE/CVF Conference on Computer Vision and Pattern Recognition (CVPR), pp. 8142-8151, 2021.
[D] S. Moon, M. Kim, Z. Qin, Y. Liu, and D. Kim, “Anonymization for skeleton action recognition,” Proceedings of the AAAI Conference on Artificial Intelligence, vol. 37, pp. 15028–15036, 2023.

---

### Meta-Review · Area_Chair_KHx1 · 2025-08-25

**Rating:** 7
**Confidence:** 4

**Strengths:**

1) The paper has a clear position: This position paper takes the stance that human pose estimation cannot be called privacy-preserving unless privacy is explicitly measured. The authors argue that current practice in the field pays almost exclusive attention to accuracy, while ignoring significant risks tied to input modalities, pose representations, and system architectures.

2) The paper provides actionable suggestions: They suggest using existing regulatory frameworks and propose concrete though preliminary indicators, such as modality scoring and re-identification testing, to bring privacy into evaluation standards.

3) The paper is well written, timely, and highly relevant.

**Weaknesses:**

1) One reviewer raised the fair concern that there is no empirical demonstration that pose outputs themselves leak private information. That is true, but for a position paper the lack of such experiments is not disqualifying: the main value lies in reframing how the community thinks about evaluation.

2) Another reviewer pointed out that the legal framing relies heavily on European and Asian regulations; including US or international frameworks would strengthen the work, though this does not undermine the core argument.

**Questions:**

-

**Ethics:**

The paper raises no ethical concerns.

**Thoroughness:**

3

---

### Decision · Program_Chairs · 2025-09-26

Accept